# Characterization of *Streptomyces sporangiiformans* sp. nov., a Novel Soil Actinomycete with Antibacterial Activity against *Ralstonia solanacearum*

**DOI:** 10.3390/microorganisms7090360

**Published:** 2019-09-17

**Authors:** Junwei Zhao, Liyuan Han, Mingying Yu, Peng Cao, Dongmei Li, Xiaowei Guo, Yongqiang Liu, Xiangjing Wang, Wensheng Xiang

**Affiliations:** 1Key Laboratory of Agricultural Microbiology of Heilongjiang Province, Northeast Agricultural University, No. 59 Mucai Street, Xiangfang District, Harbin 150030, China; guyan2080@126.com (J.Z.); 1770974118@qq.com (L.H.); 3519175610@qq.com (M.Y.); 309559770@qq.com (P.C.); 1925802035@qq.com (D.L.); 378780974@qq.com (X.G.); 616660306@qq.com (Y.L.); 2State Key Laboratory for Biology of Plant Diseases and Insect Pests, Institute of Plant Protection, Chinese Academy of Agricultural Sciences, Beijing 100193, China

**Keywords:** *Streptomyces sporangiiformans* sp. nov., antibacterial activity, multilocus sequence analysis, *Ralstonia solanacearum*

## Abstract

*Ralstonia solanacearum* is a major phytopathogenic bacterium that attacks many crops and other plants around the world. In this study, a novel actinomycete, designated strain NEAU-SSA 1^T^, which exhibited antibacterial activity against *Ralstonia solanacearum*, was isolated from soil collected from Mount Song and characterized using a polyphasic approach. Morphological and chemotaxonomic characteristics of the strain coincided with those of the genus *Streptomyces*. The 16S rRNA gene sequence analysis showed that the isolate was most closely related to *Streptomyces aureoverticillatus* JCM 4347^T^ (97.9%). Phylogenetic analysis based on 16S rRNA gene sequences indicated that the strain formed a cluster with *Streptomyces vastus* JCM4524^T^ (97.4%), *S. cinereus* DSM43033^T^ (97.2%), *S. xiangluensis* NEAU-LA29^T^ (97.1%) and *S. flaveus* JCM3035^T^ (97.1%). The cell wall contained *LL*-diaminopimelic acid and the whole-cell hydrolysates were ribose, mannose and galactose. The polar lipids were diphosphatidylglycerol (DPG), phosphatidylethanolamine (PE), hydroxy-phosphatidylethanolamine (OH-PE), phosphatidylinositol (PI), two phosphatidylinositol mannosides (PIMs) and an unidentified phospholipid (PL). The menaquinones were MK-9(H_4_), MK-9(H_6_), and MK-9(H_8_). The major fatty acids were *iso*-C_17:0_, C_16:0_ and C_17:1_ ω9c. The DNA G+C content was 69.9 mol %. However, multilocus sequence analysis (MLSA) based on five other house-keeping genes (*atp*D, *gyr*B, *rec*A, *rpo*B, and *trp*B), DNA–DNA relatedness, and physiological and biochemical data showed that the strain could be distinguished from its closest relatives. Therefore, it is proposed that strain NEAU-SSA 1^T^ should be classified as representatives of a novel species of the genus *Streptomyces*, for which the name *Streptomyces sporangiiformans* sp. nov. is proposed. The type strain is NEAU-SSA 1^T^ (=CCTCC AA 2017028^T^ = DSM 105692^T^).

## 1. Introduction

*Ralstonia solanacearum* is the causal agent of bacterial wilt, one of the most devastating plant pathogenic bacteria around the world [1], which has an unusually wide host range, infecting over 200 plant species [2], including many important agricultural crops such as potato, tomato, banana and pepper. Even though different approaches have been developed to control bacterial wilt, we still lack an efficient and environmentally friendly control measure for most of the host crops [3]. Therefore, the search and discovery of novel, environmentally friendly, commercially significant, naturally bioactive compounds are in demand to control this disease at present.

The actinobacteria are known to produce biologically active secondary metabolites, including antibiotics, enzymes, enzyme inhibitors, antitumour agents and antibacterial compounds [4,5,6]. The genus *Streptomyces*, within the family *Streptomycetaceae*, is the largest genus of the phylum *Actinobacteria*, first proposed by Waksman and Henrici (1943) [7] and currently encompasses more than 800 species with valid published names (http://www.bacterio.net/streptomyces.html), which are widely distributed in soils throughout the world. Therefore, members of novel *Streptomyces* species are in demand as sources of novel, environmentally friendly, commercially significant, naturally bioactive compounds [8,9]. During our search for antagonistic actinobacteria from soil in Mount Song, an aerobic actinomycete, strain NEAU-SSA 1^T^ with inhibitory activity against phytopathogenic bacterium *Ralstonia solanacearum* was isolated and subjected to the polyphasic taxonomy analysis. Results demonstrated that the strain represents a novel species of the genus *Streptomyces*, for which the name *Streptomyces sporangiiformans* sp. nov. is proposed.

## 2. Materials and Methods

### 2.1. Isolation of Actinomycete Strain

Strain NEAU-SSA 1^T^ was isolated from soil collected from Mount Song (34°29′ N, 113°2′ E), Dengfeng, Henan Province, China. The soil sample was air-dried at room temperature for 14 days before isolation for actinomycetes. After drying, the soil sample was ground into powder and then suspended in sterile distilled water, followed by a standard serial dilution technique. The diluted soil suspension was spread on humic acid-vitamin agar (HV) [10] supplemented with cycloheximide (50 mg L^−1^) and nalidixic acid (20 mg L^−1^). After 28 days of aerobic incubation at 28 °C, colonies were transferred and purified on the International *Streptomyces* Project (ISP) medium 3 [11], and maintained as glycerol suspensions (20%, *v*/*v*) at −80 °C for long-term preservation.

### 2.2. Morphological and Physiological and Biochemical Characteristics of NEAU-SSA 1^T^

Gram staining was carried out by using the standard Gram stain, and morphological characteristics were observed using light microscopy (Nikon ECLIPSE E200, Nikon Corporation, Tokyo, Japan) and scanning electron microscopy (Hitachi SU8010, Hitachi Co., Tokyo, Japan) using cultures grown on ISP 3 agar at 28 °C for 6 weeks. Samples for scanning electron microscopy were prepared as described by Jin et al. [12]. Cultural characteristics were determined on the ISP 1 agar [11], ISP media 2–7 [8], Czapek’s agar [13], Bennett’s agar [14], and Nutrient agar [15] after 14 days at 28 °C. Color determination was done with color chips from the ISCC-NBS (Inter-Society Color Council-National Bureau of Standards) color charts [16]. Growth at different temperatures (10, 15, 20, 25, 28, 32, 35, 40, 45, and 50 °C) was determined on ISP 3 medium after incubation for 14 days. Growth tests for pH range (pH 4.0–12.0, at intervals of 1.0 pH unit) and NaCl tolerance (0, 1, 2, 3, 4, 5, 6, 7, 8, 9, 10, 15, and 20%, *w*/*v*) were tested in GY (Glucose-Yeast extract) medium [17] at 28 °C for 14 days on a rotary shaker. The buffer systems were: pH 4.0–5.0, 0.1 M citric acid/0.1 M sodium citrate; pH 6.0–8.0, 0.1 M KH_2_PO_4_/0.1 M NaOH; pH 9.0–10.0, 0.1 M NaHCO_3_/0.1 M Na_2_CO_3_; and pH 11.0–12.0, 0.2 M KH_2_PO_4_/0.1 M NaOH. Hydrolysis of Tweens (20, 40, and 80) and production of urease were tested as described by Smibert and Krieg [18]. The utilization of sole carbon and nitrogen sources, decomposition of cellulose, hydrolysis of starch and aesculin, reduction of nitrate, coagulation and peptonization of milk, liquefaction of gelatin, and production of H_2_S were examined as described previously [19,20].

### 2.3. Chemotaxonomic Analysis of NEAU-SSA 1^T^

Biomass for chemotaxonomic studies was prepared by growing the organisms in GY medium in shake flasks at 28 °C for 5 days. Cells were harvested using centrifugation, washed with distilled water, and freeze-dried. The isomer of diaminopimelic acid (DPA) in the cell wall hydrolysates was derivatized and analyzed using an HPLC (High Performance Liquid Chromatography) method [21] with an Agilent TC-C_18_ Column (250 × 4.6 mm i.d. 5 µm; Agilent Technologies, Santa Clara, CA, USA) that had a mobile phase consisting of acetonitrile: 0.05 mol L^−1^ phosphate buffer pH 7.2 (15:85, *v*/*v*) at a flow rate of 0.5 mL min^−1^. The peak detection used an Agilent G1321A fluorescence detector (Agilent Technologies, Santa Clara, CA, USA) with a 365 nm excitation and 455 nm longpass emission filters. The whole-cell sugars were analyzed according to the procedures developed by Lechevalier and Lechevalier [22]. The polar lipids were examined using two-dimensional TLC (Thin-Layer Chromatography) and identified using the method of Minnikin et al. [23]. Menaquinones were extracted from the freeze-dried biomass and purified according to Collins [24]. Extracts were analyzed using a HPLC-UV method [25] with an Agilent Extend-C_18_ Column (150 × 4.6 mm, i.d. 5 µm; Agilent Technologies, Santa Clara, CA, USA) at 270 nm. The mobile phase was acetonitrile-*iso*-propyl alcohol (60:40, *v*/*v*). To determine cellular fatty acid compositions, the strain NEAU-SSA 1^T^ was cultivated in GY medium in shake flasks at 28 °C for 4 days. Fatty acid methyl esters were extracted from the biomass as described by Gao et al. [26] and analyzed using GC-MS according to the method of Xiang et al. [27].

### 2.4. Phylogenetic Analysis of NEAU-SSA 1^T^

For DNA extraction, strain NEAU-SSA 1^T^ was cultured in GY medium for 3 days to the early stationary phase and harvested using centrifugation. The chromosomal DNA was extracted according to the method of sodium dodecyl sulfate (SDS)-based DNA extraction [28]. PCR amplification of the 16S rRNA gene sequence was carried out using the universal bacterial primers 27F (5′-AGAGTTTGATCCTGGCTCAG-3´) and 1541R (5´-AAGGAGGTGATCCAGCC-3´) under conditions described previously [29,30]. The PCR product was purified and cloned into the vector pMD19-T (Takara) and sequenced using an Applied Biosystems DNA sequencer (model 3730XL, Applied Biosystems Inc., Foster City, California, USA). The almost complete 16S rRNA gene sequence of strain NEAU-SSA 1^T^ (1412bp) was obtained and compared with type strains available at the EzBioCloud server (https://www.ezbiocloud.net/), retrieved using NCBI BLAST (National Center for Biotechnology Information, Basic Local Alignment Search Tool; https://blast.ncbi.nlm.nih.gov/Blast.cgi;) and then submitted to the GenBank database. Phylogenetic trees were constructed based on the 16S rRNA gene sequences of strain NEAU-SSA 1^T^ and related reference species. Sequences were multiply aligned in Molecular Evolutionary Genetics Analysis (MEGA) software version 7.0 using the Clustal W algorithm and trimmed manually where necessary. Phylogenetic trees were constructed with neighbor-joining [31] and maximum likelihood [32] algorithms using MEGA [33]. The stability of the topology of the phylogenetic tree was assessed using the bootstrap method with 1000 repetitions [34]. A distance matrix was generated using Kimura’s two-parameter model [35]. All positions containing gaps and missing data were eliminated from the dataset (complete deletion option). 16S rRNA gene sequence similarities between strains were calculated on the basis of pairwise alignment using the EzBioCloud server [36]. To further clarify the affiliation of strain NEAU-SSA 1^T^ to its closely related strains, phylogenetic relationships of the strain NEAU-SSA 1^T^ were also confirmed using sequences of five individual housekeeping genes (*atp*D, g*yr*B, *rec*A, *rpo*B, and *trp*B) for core-genome analysis. The sequences of NEAU-SSA 1^T^ and its related strains were obtained from the genomes or GenBank/EMBL/DDBJ (European Molecular Biology Laboratory/DNA Data Bank of Japan). GenBank accession numbers of the sequences used are given in Table 1. The sequences of each locus were aligned using MEGA 7.0 software and trimmed manually at the same position before being used for further analysis. Trimmed sequences of the five housekeeping genes were concatenated head-to-tail in-frame in the order *atp*D-*gyr*B-*rec*A-*rpo*B-*trp*B. Phylogenetic analysis was performed as described above. Genome mining for bioactive secondary metabolites was performed using “antibiotics and secondary metabolite analysis shell” (antiSMASH) version 4.0 [37].

### 2.5. Draft Genome Sequencing and Assembly of NEAU-SSA 1^T^

For draft genome sequencing and assembly, the genomic DNA of strain NEAU-SSA 1^T^ was extracted using the method of SDS-based DNA extraction [28]. The harvested DNA was detected using agarose gel electrophoresis and quantified using Qubit^®^ 2.0 Fluorometer (Thermo Scientific). Whole-genome sequencing was performed on the Illumina HiSeq PE150 (Illumina, San Diego, CA, USA) platform. A-tailed, ligated to paired-end adaptors, and PCR amplified samples with a 350 bp insert were used for the library construction at the Beijing Novogene Bioinformatics Technology Co., Ltd. Illumina PCR adapter reads and low-quality reads from the paired-end were filtered using a quality control step using our own compling pipeline. All good-quality paired reads were assembled using the SOAP (Short Oligonucleotide Alignment Program) denovo [38,39] (https://github.com/aquaskyline) into a number of scaffolds. Then, the filter reads were handled by the next step of the gap-closing.

### 2.6. DNA–DNA Relatedness Tests

Because of a lacking number of genome sequences of *Streptomyces aureoverticillatus* JCM4347^T^, *Streptomyces vastus* JCM4524^T^, *S. cinereus* DSM43033^T^, and *S. xiangluensis* NEAU-LA29^T^, DNA–DNA relatedness tests between strain NEAU-SSA 1^T^ and those strains were carried out as described by De Ley et al. [40] under consideration of the modifications described by Huss et al. [41], using a model Cary 100 Bio UV/VIS-spectrophotometer (Hitachi U-3900, Hitachi Co., Tokyo, Japan) equipped with a Peltier-thermostatted 6 × 6 multicell changer and a temperature controller with in situ temperature probe (Varian). The genomic DNAs of strain NEAU-SSA 1^T^ and its closely related species—*S. aureoverticillatus* JCM4347^T^, *S. vastus* JCM4524^T^, *S. cinereus* DSM43033^T^, and *S. xiangluensis* NEAU-LA29^T^—were extracted using the method of SDS-based DNA extraction [28]. The concentration and purity of these DNA samples were determined by measuring the optical density (OD) at 260, 280, and 230 nm. The DNA samples used for hybridization were diluted to OD_260_ around 1.0 using 0.1 × SSC (saline sodium citrate buffer), then sheared using a JY92-II ultrasonic cell disruptor (ultrasonic time 3 s, interval time 4 s, 90 times; Ningbo Scientz Biotechnology Co., Ltd, Ningbo, China). The DNA renaturation rates were determined in 2 × SSC at 70 °C. The experiments were performed with three replications and the DNA–DNA relatedness value was expressed as a mean of the three values. Several genomic metrics are now available to distinguish between orthologous genes of closely related prokaryotes, including the calculation of average nucleotide identity (ANI) and digital DNA–DNA hybridization (dDDH) values [42,43]. In the present study, ANI and dDDH values were determined from the genomes of strain NEAU-SSA 1^T^ and *S. flaveus* JCM3035^T^ (JOCU00000000) using the ortho-ANIu algorithm from Ezbiotaxon and the genome-to-genome distance calculator (GGDC 2.0) at http://ggdc.dsmz.de.

### 2.7. In Vitro Antibacterial Activity Test

The antibacterial activity of strain NEAU-SSA 1^T^ against two pathogenic bacteria (*Micrococcus luteus* and *Ralstonia solanacearum*) was evaluated using the agar well diffusion method [44] with the cultures growth on ISP 3 medium at 28 °C for four weeks as follows: All the spores and mycelia were collected from one ISP 3 plate (diameter, 9mm) and then extracted using 1 mL methanol with an ultrasonic step (300 W, 30–60 min). Afterwards, 200 µL methanol extract or methanol was added to the agar well, and methanol was used as the control. To further investigate the antibacterial components produced by NEAU-SSA 1^T^, the strain was cultured in tryptone-glucose-soluble starch-yeast extract medium (tryptone 0.2%, glucose 1%, soluble starch 0.5%, yeast extract 0.2%, NaCl 0.4%, K_2_HPO_4_ 0.05%, MgSO_4_.7H_2_O 0.05%, CaCO_3_ 0.2%, *w*/*v*, pH 7.0–7.4), and the inhibitory activity was tested. Briefly, strain NEAU-SSA 1^T^ was inoculated into MB medium and incubated at 28 °C for seven days in a rotary shaker. The supernatant (100 mL for this study) was obtained via centrifugation at 8000 rpm and 4 °C for 10 min and subsequently extracted by using an equal volume of ethyl acetate. Then, the extract was dried in a rotary evaporator at 40 °C and eluted with proper volume methanol (1 mL used in this study). The cell precipitate was extracted with an equal volume of methanol and also condensed as above. After that, the antibacterial activity was evaluated using the agar well diffusion method, and each well contained 200 µL of the methanol extract. To examine the effect of temperature on antibacterial activity, the ten-fold dilution methanol extract was placed in a water bath at 40, 60, 80, and 100 °C for 30 min, and then cooled to room temperature. The antibacterial activity was evaluated using the agar well diffusion method.

## 3. Result and Discussion

### 3.1. Polyphasic Taxonomic Characterization of NEAU-SSA 1^T^

The morphological characteristics of strain NEAU-SSA 1^T^ showed that the strain had the typical characteristics of the genus *Streptomyces*. Observation of 6-week cultures of strain NEAU-SSA 1^T^ grown on ISP 3 medium revealed that it formed well-developed, branched substrate hyphae and aerial mycelia. Sporangia consisted of cylindrical, and rough-surfaced spores (0.6–0.8 μm × 0.9–1.6 μm) were produced on aerial mycelia, but spore chains were not observed (Figure 1). Strain NEAU-SSA 1^T^ exhibited good growth on ISP 3, ISP 4, ISP 7, and Nutrient agar media; moderate growth on ISP 1, ISP 2, ISP 5, ISP 6, and Czapek’s agar media; and poor growth on Bennett’s agar medium. The cultural characteristics of strain NEAU-SSA 1^T^ is shown in Appendix A. Strain NEAU-SSA 1^T^ grew well between pH 6.0 and 11.0, with an optimum pH of 7.0. The range of temperature of the strain was determined to be 15–45 °C, with the optimum growth temperature being 28 °C. The strain grew in the presence of 0–6% NaCl (*w*/*v*) with an optimal level of 0–1% (*w*/*v*). Detailed physiological characteristics are presented in the species description (Table 2 and Appendix A).

Chemotaxonomic analyses revealed that strain NEAU-SSA 1^T^ exhibited characteristics that are typical of representatives of the genus *Streptomyces*. The strain was found to contain *LL*-diaminopimelic acid as diamino acid. The whole-cell hydrolysates of the strain were determined to contain ribose, mannose, and galactose. The menaquinones of strain NEAU-SSA 1^T^ were MK-9(H_4_) (29.5%), MK-9(H_6_) (41.2%), and MK-9(H_8_) (29.4%). The cellular fatty acid profile of strain NEAU-SSA 1^T^ was composed of *iso*-C_17:0_ (30.9%), C_16:0_ (26.4%), C_17:1_ω9c (19.9%), C_15:0_ (7.8%), C_17:0_ (4.4%), C_14:0_ (3.3%), *iso*-C_16:0_ (1.7%), *anteiso*-C_15:0_ (1.7%), C_18:1_ω9c (1.7%), C_16:0_ 1-OH (1.2%), and *iso*-C_18:0_ (1.1%). The polar lipids of the strain consisted of diphosphatidylglycerol (DPG), phosphatidylethanolamine (PE), hydroxy-phosphatidylethanolamine (OH-PE), phosphatidylinositol (PI), two phosphatidylinositol mannosides (PIMs), and an unidentified phospholipid (PL) (Appendix A). All the chemotaxonomic data are consistent with the assignment of strain NEAU-SSA 1^T^ to the genus *Streptomyces*.

Sequence analysis of the 16S rRNA gene showed that strain NEAU-SSA 1^T^ were affiliated with the genus *Streptomyces* and most closely related to *S. aureoverticillatus* JCM 4347^T^ (97.9%). Phylogenetic analysis based on 16S rRNA gene sequences indicated that the strain formed a cluster with *S. vastus* JCM4524^T^ (97.4%), *S. cinereus* DSM43033^T^ (97.2%), *S. xiangluensis* NEAU-LA29^T^ (97.1%), and *S. flaveus* JCM3035^T^ (97.1%) in the neighbor-joining tree (Figure 2), a relationship also recovered by the maximum-likelihood algorithm (Appendix A). Phylogenetic trees based on the neighbor-joining and maximum-likelihood algorithms were constructed from the concatenated sequence alignment of the five housekeeping genes (Figure 3 and Appendix A), and had the same topology as the 16S rRNA gene tree. Moreover, pairwise distances calculated for NEAU-SSA 1^T^ and the related species using the concatenated sequences of *atp*D-*gyr*B-*rec*A-*rpo*B-*trp*B were well above 0.007 (Appendix A), which is considered to be the threshold for species determination by Rong et al. [46]. DNA–DNA hybridization was employed to further clarify the relatedness between the strain and *S. aureoverticillatus* JCM 4347^T^, *S. vastus* JCM4524^T^, *S. cinereus* DSM 43033^T^, and *S. xiangluensis* NEAU-LA29^T^. Results showed that strain NEAU-SSA 1^T^ shared DNA–DNA relatedness of 37.1 ± 3.4% with *S. aureoverticillatus* JCM 4347^T^, 35.4 ± 4.3% with *S. vastus* JCM 4524^T^, 33.1 ± 4.1% with *S. cinereus* DSM 43033^T^, and 29.0 ± 4.9% with *S. xiangluensis* NEAU-LA29^T^. Digital DNA–DNA hybridization was employed to clarify the relatedness between strain NEAU-SSA 1^T^ and *S. flaveus* JCM 3035^T^. The level of digital DNA–DNA hybridization between them was 24.9 ± 2.4%. These five values are all below the threshold value of 70% recommended by Wayne et al. [47] for assigning strains to the same genomic species. Similarly, a low ANI value of 80.99% was found between strain NEAU-SSA 1^T^ and *S. flaveus* JCM 3035^T^, a result well below the threshold used to delineate prokaryote species [48,49].

The assembled genome sequence of strain NEAU-SSA 1^T^ was found to be 10,364,704 bp long and composed of 352 contigs with an N50 of 59,982 bp, a DNA G+C content of 69.9 mol % and a coverage of 200x. It was deposited into GenBank under the accession number VCHX00000000. The 16S rRNA gene sequence from the whole genome sequence shared a 100% similarity with that from PCR sequencing, suggesting that the genome sequence was not contaminated. Detailed genomic information is presented in the Appendix A.

Comparison of phenotypic characteristics between strain NEAU-SSA 1^T^ and its closely related species—*S. aureoverticillatus* JCM 4347^T^, *S. vastus* JCM4524^T^, *S. cinereus* DSM 43033^T^, *S. xiangluensis* NEAU-LA29^T^, and *S. flaveus* JCM 3035^T^—was performed to differentiate these strains (Table 2). Differential cultural characteristics included: NaCl tolerance of the strain was up to 5.0%, which is lower than that of *S. aureoverticillatus* JCM 4347^T^ (15%) and *S. flaveus* JCM 3035^T^ (7%); and the strain could grow at pH 11.0, while *S. vastus* JCM4524^T^, *S. cinereus* DSM 43033^T^, and *S. xiangluensis* NEAU-LA29^T^ could not. Other phenotypic differences included the production of H_2_S; decomposition of cellulose; liquefaction of gelatin; growth temperature; hydrolysis of Tweens (20, 40, and 80); and utilization of L-arabinose, D-galactose, D-fructose, D-maltose, lactose, L-rhamnose, D-ribose, D-sorbitol, D-mannose, raffinose, D-xylose, *myo*-inositol, L-glutamine, glycine, L-threonine, L-tyrosine, L-serine, L-proline, L-asparagine, and L-arginine.

On the basis of morphological, physiological, chemotaxonomic, and phylogenetic results, strain NEAU-SSA 1^T^ is considered to represent a novel species within the genus *Streptomyces*, for which the name *Streptomyces sporangiiformans* is proposed.

### 3.2. Description of Streptomyces sporangiiformans sp. nov.

*Streptomyces sporangiiformans* (spo.ran.gi.i.for’mans. N.L. neut. n. sporangium; L. pres. part. f*ormans* forming; N.L. part. adj. *sporangiiformans* forming sporangia).

Gram-stain-positive, aerobic actinomycete that formed well-developed, branched substrate hyphae and aerial mycelia. Sporangia consisted of cylindrical and rough surfaced spores (0.6–0.8 μm × 0.9–1.6 μm) were produced on aerial mycelia, but spore chains were not observed. Good growth on ISP 3, ISP 4, ISP 7, and Nutrient agar media; moderate growth on ISP 1, ISP 2, ISP 5, ISP 6, and Czapek’s agar media; and poor growth on Bennett’s agar medium. Growth occurred at pH values between 6.0 and 11.0, the optimum being pH 7.0. Tolerates up to 6.0% NaCl and grows optimally in 0–1% (*w*/*v*) NaCl. Growth was observed at temperatures between 15 and 45 °C, with an optimum temperature of 28 °C. Positive for decomposition of Tweens (40 and 80) and cellulose, hydrolysis of aesculin and starch, liquefaction of gelatin and production of urease; and negative for coagulation and peptonization of milk, hydrolysis of Tween 20, production of H_2_S, and reduction of nitrate. D-fructose, D-galactose, D-glucose, inositol, lactose, D-maltose, D-mannose, D-raffinose, L-rhamnose, and D-sucrose were utilized as sole carbon sources, but not L-arabinose, dulcitol, D-ribose, D-sorbitol, or D-xylose. L-alanine, L-arginine, L-asparagine, L-aspartic acid, creatine, L-glutamic acid, L-glutamine, L-proline, L-serine, and L-threonine were utilized as sole nitrogen sources, but not glycine or L-tyrosine. Cell wall contained *LL*-diaminopimelic acid and the whole-cell hydrolysates were ribose, mannose, and galactose. The polar lipids contained diphosphatidylglycerol (DPG), phosphatidylethanolamine (PE), hydroxy-phosphatidylethanolamine (OH-PE), phosphatidylinositol (PI), two phosphatidylinositol mannosides (PIMs), and an unidentified phospholipid (PL). The menaquinones were MK-9(H_4_), MK-9(H_6_), and MK-9(H_8_). Major fatty acids were *iso*-C_17:0_, C_16:0_, and C_17:1_ω9c.

The type strain was NEAU-SSA 1^T^ (=CCTCC AA 2017028^T^ = DSM 105692^T^), isolated from soil collected from Mount Song, Dengfeng, Henan Province, China. The DNA G+C content of the type strain was 69.9 mol %, calculated from the assembly for the draft genome sequence. The GenBank/EMBL/DDBJ accession number for the 16S rRNA gene sequence of strain NEAU-SSA 1^T^ is MH842151. This Whole Genome Shotgun project has been deposited at DDBJ/ENA/GenBank under the accession VCHX00000000. The version described in this paper is version VCHX00000000.2.

### 3.3. Antibacterial Activity of NEAU-SSA 1^T^ against Ralstonia solanacearum

Strain NEAU-SSA 1^T^ exhibited antibacterial activity against *Ralstonia solanacearum* with inhibitory zone diameters of 23 mm (Figure 4a). However, no inhibitory effect on the growth of *Micrococcus luteus* (Figure 4b) was observed. Comparison of the antibacterial activity of the extract of the supernatant with that of the cell pellet suggested that the antibacterial substances of strain NEAU-SSA 1^T^ were in both the supernatant and cell pellet since the extracts all showed inhibition of the growth of *Ralstonia solanacearum* with the inhibitory zone diameters of 31.5 and 26.4 mm, respectively (Figure 5a,b). The antibacterial substances in the supernatant were stable after they were placed in a water bath at 40 and 60 °C for 30 min, while they did not show antibacterial activity after 80 and 100 °C bath (Figure 6a), which indicated that they were sensitive to temperature. In contrast, the antibacterial substances in the cell pellet were insensitive to temperature (Figure 6b), which demonstrated that the antibacterial substances in the supernatant and cell pellet were different. The antiSMASH analysis led to the identification of 49 gene clusters, including 24 gene clusters that showed very low similarity to the known gene clusters of mediomycin A, cremimycin, primycin, ibomycin, naphthomycin, lasalocid, informatipeptin, polyoxypeptin, kutznerides, anisomycin, paulomycin, himastatin, desotamide, nystatin, tiacumicin B, oxazolomycin, and 4-Z-annimycin. Therefore, the relationships between the corresponding secondary metabolites produced by NEAU-SSA 1^T^ and the antibacterial activity are still ambiguous. *Streptomyces* are well known as important biological resources for their biologically active secondary metabolites, which play important roles in protecting plants against pathogens [50]. Strain NEAU-SSA 1^T^, which shows a stronger antibacterial activity against *Ralstonia solanacearum*, is a novel species of the genus *Streptomyces*, and possesses 24 lower similarity gene clusters. Therefore, it is interesting and significant to isolate and identify the secondary metabolites of the strain in further studies.

## 4. Conclusions

A novel strain NEAU-SSA 1^T^ that exhibited antibacterial activity against *Ralstonia solanacearum* was isolated from a soil sample. Morphological features, phylogenetic analysis based on 16S rRNA gene sequences, and multilocus sequence analysis based on five other house-keeping genes (*atp*D, *gyr*B, *rec*A, *rpo*B, and *trp*B) suggested that strain NEAU-SSA 1^T^ belonged to the genus *Streptomyces*. Physiology and biochemical characteristics, together with DDH relatedness values and ANI values, clearly indicated that strain NEAU-SSA 1^T^ could be differentiated from the closely related strains *S. aureoverticillatus* JCM 4347^T^, *S. vastus* JCM 4524^T^, *S. cinereus* DSM 43033^T^, *S. xiangluensis* NEAU-LA29^T^, and *S. flaveus* JCM 3035^T^. Based on the polyphasic analysis, it is proposed that strain NEAU-SSA 1^T^ should be classified as representatives of a novel species of the genus *Streptomyces*, for which the name *Streptomyces sporangiiformans* sp. nov. is proposed. The type strain is NEAU-SSA 1^T^ (=CCTCC AA 2017028^T^ = DSM 105692^T^).

## Figures and Tables

**Figure 1 microorganisms-07-00360-f001:**
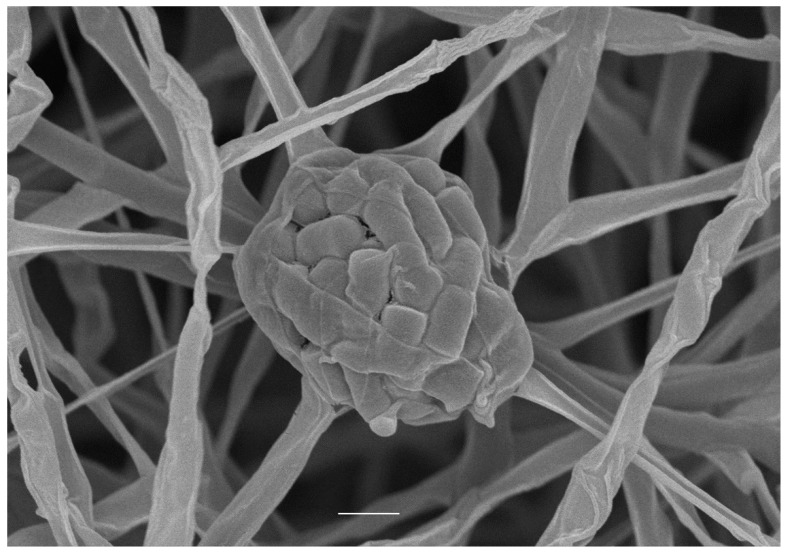
Scanning electron micrograph of strain NEAU-SSA 1^T^ grown on ISP 3 agar for 6 weeks at 28 °C; Scale bar represents 1 μm.

**Figure 2 microorganisms-07-00360-f002:**
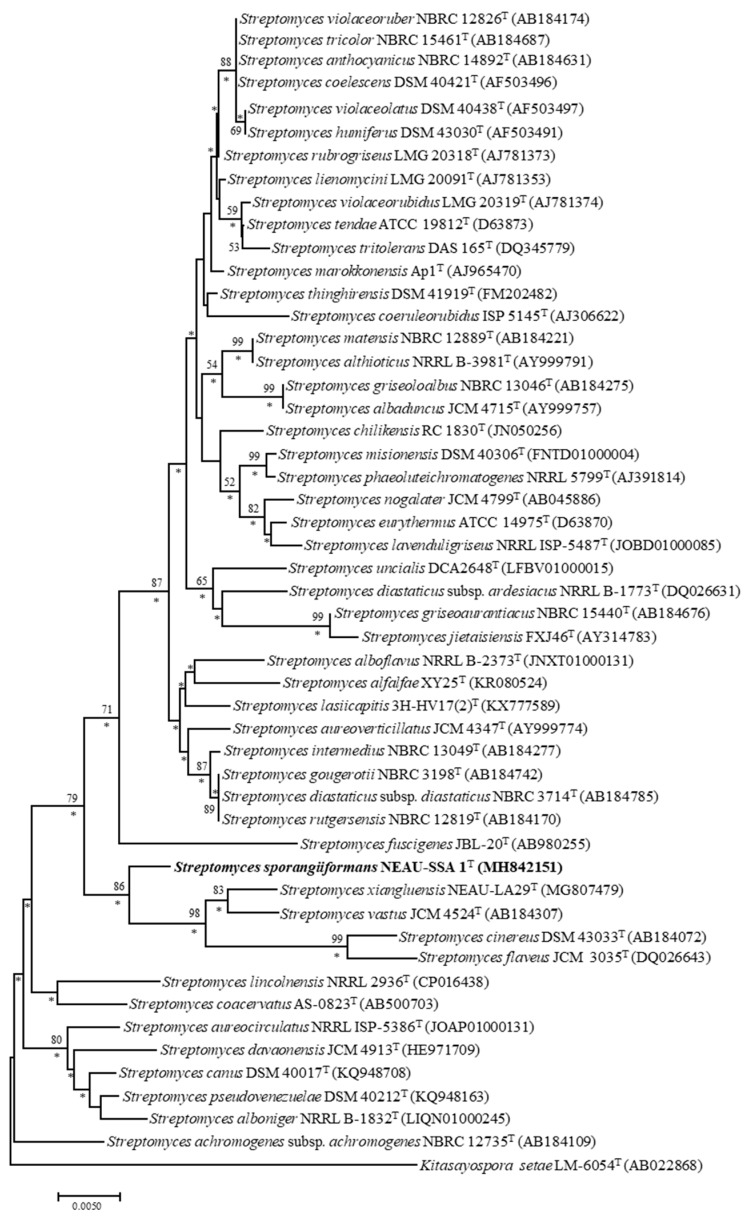
Neighbor-joining tree showing the phylogenetic position of strain NEAU-SSA 1^T^ (1412 bp) and the related species of the genus *Streptomyces* based on 16S rRNA gene sequences. The out-group used was *Kitasatospora setae* LM-6054^T^. Only bootstrap values above 50% (percentages of 1000 replications) are indicated. Asterisks indicate branches also recovered in the maximum-likelihood tree. Scale bar represents 0.005 nucleotide substitutions per site.

**Figure 3 microorganisms-07-00360-f003:**
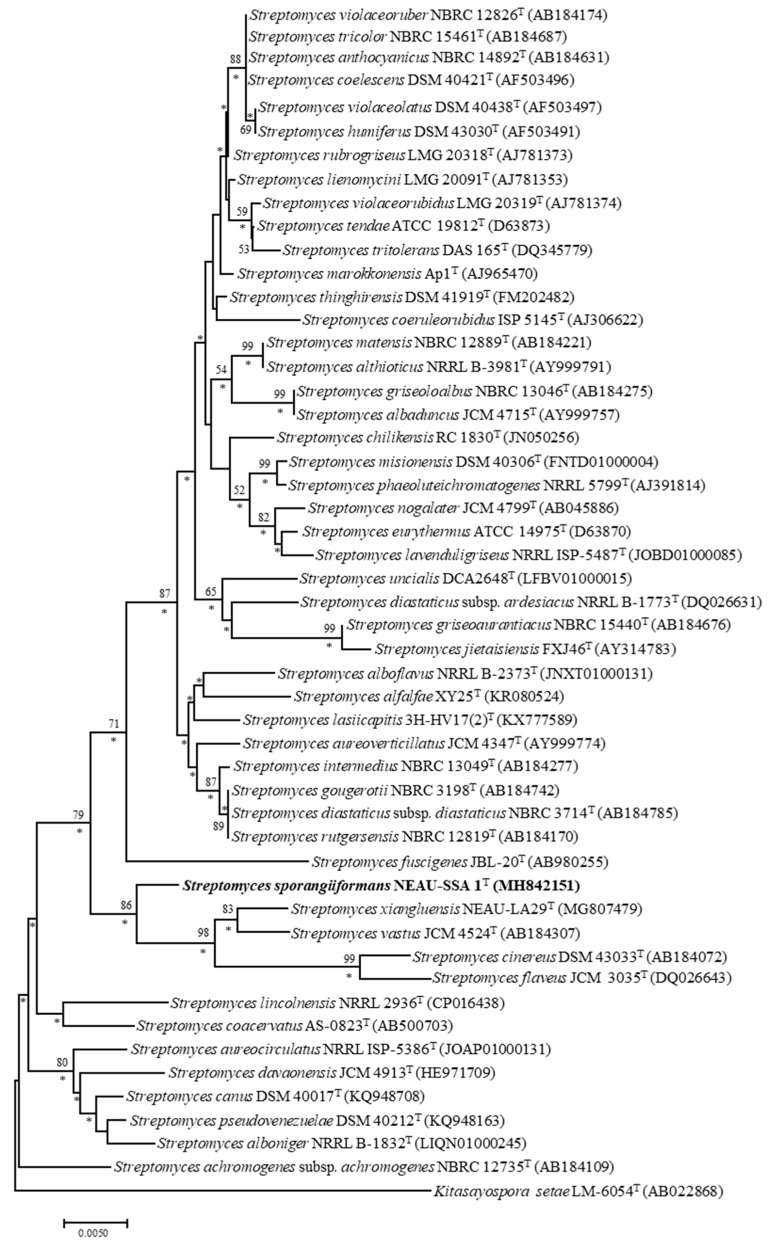
Neighbor-joining tree based on MLSA analysis of the concatenated partial sequences from five housekeeping genes (*atp*D, *gyr*B, *rec*A, *rpo*B, and *trp*B) of isolate NEAU-SSA 1^T^ (in bold) and related taxa. Only bootstrap values above 50% (percentages of 1000 replications) are indicated. *Kitasatospora setae* LM-6054^T^ was used as an out-group. Asterisks indicate branches also recovered in the maximum-likelihood tree. Scale bar represents 0.02 nucleotide substitutions per site.

**Figure 4 microorganisms-07-00360-f004:**
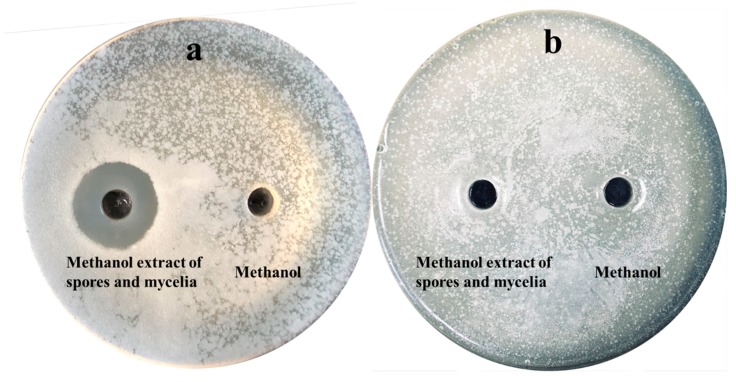
The antibacterial activity of strain NEAU-SSA 1^T^ against *Ralstonia solanacearum* (**a**) and *Micrococcus luteus* (**b**).

**Figure 5 microorganisms-07-00360-f005:**
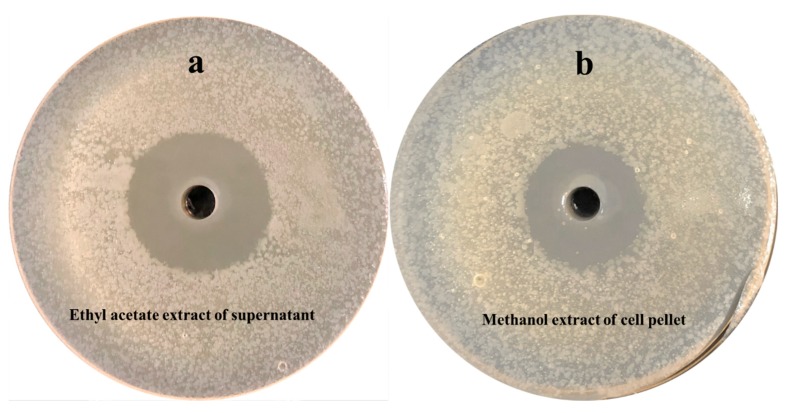
The antibacterial activity of the extract of the supernatant (**a**) and cell pellet of strain NEAU-SSA 1^T^ (**b**) against *Ralstonia solanacearum*.

**Figure 6 microorganisms-07-00360-f006:**
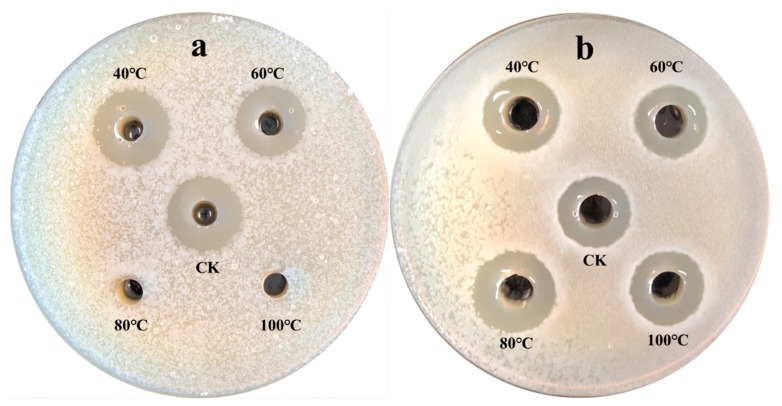
The effect of temperature on the antibacterial activity of the extract of the supernatant (**a**) and cell pellet of strain NEAU-SSA 1^T^ (**b**) against *Ralstonia solanacearum*.

**Table 1 microorganisms-07-00360-t001:** GenBank Accession Numbers of the Sequences Used in MLSA.

Strain	Type Strain	Whole Genome	*atp*D	*gyr*B	*rec*A	*rpo*B	*trp*B
*Streptomyces sporangiiformans*	NEAU-SSA 1^T^	VCHX00000000		–	–	–	–
*Streptomyces coelescens*	DSM 40421^T^	–	GU383344	AY508508	KT385220	GU383768	KT389192
*Streptomyces violaceolatus*	DSM 40438^T^	–	GU383347	AY508509	KT385451	GU383771	KT389418
*Streptomyces anthocyanicus*	NBRC 14892^T^	–	KT384465	KT384814	KT385162	KT388784	KT389134
*Streptomyces humiferus*	DSM 43030^T^	–	KT384598	KT384947	KT385296	KT388918	KT389267
*Streptomyces violaceoruber*	NBRC 12826^T^	CP020570	KT384751	KT385099	KT385453	KT389071	KT389420
*Streptomyces rubrogriseus*	LMG 20318^T^	BEWD00000000	KT384715	KT385065	KT385416	KT389036	KT389384
*Streptomyces tendae*	ATCC 19812^T^	–	KT384733	KT385082	KT385434	KT389053	KT389402
*Streptomyces violaceorubidus*	LMG 20319^T^	JODM00000000	–	–	–	–	–
*Streptomyces lienomycini*	LMG 20091^T^	–	KT384622	KT384971	KT385321	KT388942	KT389291
*Streptomyces diastaticus* subsp*. ardesiacus*	NRRL B-1773^T^	BEWC00000000	KT384534	KT384883	KT385231	KT388853	KT389203
*Streptomyces albaduncus*	JCM 4715^T^	–	KT384449	KT384798	KT385146	KJ996741	KT389118
*Streptomyces matensis*	NBRC 12889^T^	–	KT384637	KT384986	KT385337	KT388957	KT389306
*Streptomyces althioticus*	NRRL B-3981^T^	–	KT384460	KT384809	KT385157	KT388779	KT389129
*Streptomyces davaonensis*	JCM 4913^T^	HE971709	–	–	–	–	–
*Streptomyces canus*	DSM 40017^T^	LMWO00000000	KT384500	KT384849	KT385197	KT388819	KT389169
*Streptomyces lincolnensis*	NRRL 2936^T^	CP016438	–	–	–	–	–
*Streptomyces pseudovenezuelae*	DSM 40212^T^	LMWM00000000	KT384695	KT385045	KT385396	KT389016	KT389364
*Streptomyces xiangluensis*	NEAU-LA29^T^	–	MH291276	MH345670	MH291277	MH291275	MH291278
*Streptomyces vastus*	NBRC 13094^T^	–	KU323834	KT385093	KU975607	KT389065	KT389414
*Streptomyces cinereus*	NBRC 12247^T^	–	KT384513	KT384862	KT385210	KJ996667	KT389182
*Streptomyces flaveus*	NRRL B-16074^T^	JOCU00000000	KT384551	KT384900	KT385249	KT388870	KT389220
*Streptomyces chilikensis*	RC 1830^T^	LWCC00000000	–	–	–	–	–
*Streptomyces coeruleorubidus*	ISP 5145^T^	–	KT384528	KT384877	KT385225	KT388847	KT389197
*Streptomyces misionensis*	DSM 40306^T^	FNTD00000000	KT384647	KT384996	KT385347	KT388967	KT389316
*Streptomyces phaeoluteichromatogenes*	NRRL 5799^T^	–	KT384680	KT385030	KT385381	KT389001	KT389350
*Streptomyces tricolor*	NBRC 15461^T^	MUMF00000000	KT384741	KT385089	KT385443	KT389061	KT389410
*Streptomyces achromogenes* subsp*. achromogenes*	NBRC 12735^T^	JODT00000000	–	–	–	–	–
*Streptomyces eurythermus*	ATCC 14975^T^	–	KT384544	KT384893	KT385242	KT388863	KT389213
*Streptomyces nogalater*	JCM 4799^T^	–	KT384664	KT385014	KT385365	KT388984	KT389333
*Streptomyces jietaisiensis*	FXJ46^T^	FNAX00000000	KT384605	KT384954	KT385304	KT388925	KT389274
*Streptomyces griseoaurantiacus*	NBRC 15440^T^	AEYX00000000	–	–	–	–	–
*Streptomyces lavenduligriseus*	NRRL ISP-5487^T^	JOBD00000000	KT384620	AB072859	KT385319	KT388940	KT389289
*Streptomyces uncialis*	DCA2648^T^	LFBV00000000	–	–	–	–	–
*Streptomyces alboniger*	NRRL B-1832^T^	–	KT384455	KT384804	KT385152	KT388774	KT389124
*Streptomyces alfalfae*	XY25^T^	CP015588	–	–	–	–	–
*Streptomyces lasiicapitis*	3H-HV17(2)^T^	–	MH651782	KY229066	MH651785	MH651788	MH651791
*Streptomyces aureocirculatus*	NRRL ISP-5386^T^	JOAP00000000	KT384476	KT384825	KT385173	KT388795	KT389145
*Streptomyces aureoverticillatus*	NRRL B-3326^T^	–	KT384478	KT384827	KT385175	KT388797	KT389147
*Streptomyces alboflavus*	NRRL B-2373^T^	CP021748	–	–	–	–	–
*Streptomyces rutgersensis*	NBRC 12819^T^	–	KT384716	KT385066	KT385417	KT389037	KT389385
*Streptomyces intermedius*	NBRC 13049^T^	–	KT384602	KT384951	KT385301	KT388922	KT389271
*Streptomyces gougerotii*	NBRC 3198^T^	–	KT384572	KT384921	KT385270	KT388891	KT389241
*Streptomyces diastaticus* subsp*. diastaticus*	NBRC 3714^T^	–	KT384535	KT384884	KT385232	KT388854	KT389204
*Kitasatospora setae*	LM-6054^T^	AP010968	–	–	–	–	–

**Table 2 microorganisms-07-00360-t002:** Differential characteristics of strain NEAU-SSA 1^T^, *S. aureoverticillatus* JCM 4347^T^, *S. vastus* JCM4524^T^, *S. cinereus* DSM43033^T^, *S. xiangluensis* NEAU-LA29^T^, and *S. flaveus* JCM3035^T^.

Characteristic	1	2	3	4	5	6^a^
Decomposition of cellulose	+	–	–	–	+	ND
Production of H_2_S	–	–	–	–	–	+
Tween 20	–	+	–	–	–	ND
Tween 40	+	+	–	–	+	ND
Tween 80	+	+	–	–	+	ND
Liquefaction of gelatin	+	–	–	–	–	ND
Growth temperature (℃)	15–45	10–45	20–40	20–40	20–40	10–37
pH range for growth	6–11	5–12	6–10	6–9	6–9	ND
NaCl tolerance range (*w*/*v*, %)	0–6	0–15	0–5	0–5	0–6	0–7
Carbon source utilization						
D-fructose	+	+	–	–	+	+
D-galactose	+	+	–	–	+	+
Lactose	+	+	–	–	+	+
D-maltose	+	+	–	–	+	ND
L-rhamnose	+	+	–	+	+	+
D-ribose	–	+	–	–	–	–
D-sorbitol	–	+	–	+	–	ND
D-mannose	+	+	+	–	+	+
Raffinose	+	+	+	–	+	+
L-arabinose	–	–	–	–	–	+
D-xylose	–	–	–	–	–	+
Myo-inositol	+	+	+	–	+	+
Nitrogen source utilization						
L-glutamine	+	+	–	–	+	ND
Glycine	–	+	+	+	–	ND
L-threonine	+	+	–	–	–	+
L-tyrosine	–	–	–	+	+	ND
L-arginine	+	+	+	+	–	+
L-asparagine	+	+	+	+	–	ND
L-serine	+	+	+	+	–	+
L-proline	+	+	+	+	–	–

Strains: 1—NEAU-SSA 1^T^; 2—*S. aureoverticillatus* JCM 4347^T^; 3—*S. vastus* JCM4524^T^; 4—*S. cinereus* DSM43033^T^; 5—*S. xiangluensis* NEAU-LA29^T^; 6—*S. flaveus* JCM3035^T^. Abbreviation: +, positive; –, negative. All data are from this study except where marked. ^a^ Data from Michael Goodfellow et al. [45].

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
