# Peer review of "Characterization of *Streptomyces sporangiiformans* sp. nov., a Novel Soil Actinomycete with Antibacterial Activity against *Ralstonia solanacearum"

_microorganisms, 2019, doi:10.3390/microorganisms7090360_

Round 1
Reviewer 1 Report
The work entitled “Characterization of Streptomyces sporangiiformans sp. nov., a novel soil actinomycete with antibacterial activity against Ralstonia solanacearum”, authors are J. Zhao, L. Han, M. Yu, P. Cao, D. Li, X. Guo, Y. Liu, X. Wang, and W. Xiang, is actual and devoted to the search for novel producers of antibiotics against phytopathogenic microorganisms. The authors have isolated and identified novel Streptomyces species with high antibacterial activity against plant pathogenic bacterium Ralstonia solanacearum. In the work, various modern methods were used; and the strain Streptomyces sporangiiformans NEAU-SSA 1T was described using polyphasic approach that included analysis of genotypic (the 16S rRNA gene sequence, multilocus sequence analysis with 5 house-keeping genes, draft genome sequence, and DNA-DNA hybridization) and phenotypic (chemotaxonomy, morphology, growth substrate specificity, and growth conditions) traits.
Specific remarks
It is not clear why only the isolate designated authors as NEAU-SSA 1T was selected for analysis and identification since, apparently, many colonies of actinobacteria were isolated from the soil suspension.
There are some type mistakes: p. 3, line 4 – s in Strain should be lowercase; p. 4, line 8 – Micrococcus not Micrococus; p. 4, line 10 – “below” not “bellow”.
Please check if gene names ( atpD, gyrB, recA, rpoB, and trpB) and Latin names (especially in References) are italized everywhere in the text.
Summary. The work is recommended for publication in the journal “Microorganisms” after very minor revision.
Author Response
Dear reviewer,
Thank you for the valuable suggestions. In the following, we provide our itemized list of changes according to your suggestions.
Thank you very much for your kindness and help.
Sincerely yours,
Wensheng Xiang
It is not clear why only the isolate designated authors as NEAU-SSA 1T was selected for analysis and identification since, apparently, many colonies of actinobacteria were isolated from the soil suspension.
Reply: Because strain NEAU-SSA 1T has strong antibacterial activity against Ralstonia solanacearum and shows low 16S rRNA gene sequence to other strains of the genus Streptomyces. The strain NEAU-SSA 1T may possess new genes and produce new natural active substances. Furthermore, the strain can be a potential candidate for development of potential biocontrol agent against Ralstonia solanacearum used in organic agriculture. So the strain was selected for analysis and identification.
There are some type mistakes: p. 3, line 4 – s in Strain should be lowercase; p. 4, line 8 – Micrococcus not Micrococus; p. 4, line 10 – “below” not “bellow”.
Reply: We have revised. Please see p. 3, line 6; and p. 5, lines 23 and25;
Please check if gene names (atpD, gyrB, recA, rpoB, and trpB) and Latin names (especially in References) are italized everywhere in the text.
Reply: Thank you very much for your valuable suggestion. We have checked and revised.
Reviewer 2 Report
This manuscript characterizes a novel species of soil Actinomycete, with antibacterial activity adequate to be used against Ralstonia solanacearum.
This reviewer enjoyed reading the manuscript and has only minor comments/suggestions/queries. To facilitate the review process an annotated pdf is attached, for the authors.

Author Response
Dear Reviewer,
Thank you for the valuable suggestions. In the following, we provide our itemized list of changes according to your suggestions.
Thank you very much for your kindness and help.
Sincerely yours,
Wensheng Xiang
Page 1, line 38: Remove ‘and so on’
We have revised. Thank you. (Please see page 1, line 38)
Page 1, line 45: Remove ‘and was’ and add ‘,’; remove ‘,’ and add ‘and’
We have revised. Thank you. (Please see page 2, line 1)
Page 2, line 1: add “are” before “widely”
We have revised. Thank you. (Please see page 2, line 2)
Page 3, line 6: reference is Sambrook. It is too vague to reference the whole volume. Please indicate the precise method used
We have added the method and also the new reference. Thank you. (Please see page 3, line 8 and reference 28)
Page 3, line 15: Please give the software full name here and not below
We have revised. Thank you. (Please see page 3, lines 17-18 and 20)
Page 3, line 17: The ML reference is number 32 and not 31
We have revised. Thank you. (Please see page 3, line 20)
Page 3, line 24: Please clarify the “SDS method”
We added a reference of the method. Thank you. (Please see page 4, line 4 and reference 28)
Page 3, lines 23-31: In my view this parte needs to be in Materials and methods, in a dedicated section. This is not phylogenetic analysis.
We have revised and listed this part as a dedicated section of 2.5 (Draft Genome Sequencing and Assembly of NEAU-SSA 1T). Thank you. (Please see page 4, lines 2-11)
Page 3, line 42: remove “them”, add “those strains”
We have revised. Thank you. (Please see page 5, line 4)
Page 3, lines 46-47: What was the origin of these DNA samples? How were they obtained?
We have added the sentence ‘The genomic DNAs of strain NEAU-SSA 1T and its closely related species, S. aureoverticillatus JCM4347T, S. vastus JCM4524T, S. cinereus DSM43033T and S. xiangluensis NEAU-LA29T, were extracted with the method of SDS-based DNA extraction’ to this part of our manuscript and changed ‘the DNA samples’ to ‘these DNA samples’. Thank you. (Please see page 5, lines 7-10)
Page 4, line 3: include “(dDDH)” before “values”
We have added. Thank you. (Please see page 5, line 18)
Page 4, line 10: Change ‘was’ to ‘were’
We have revised. Thank you. (Please see page 5, line 25)
Page 4, line 13: do the authors mean “the same volume of methanol”? As it is the sentence does not read well.
We have changed the sentence to ‘Afterwards, 200 µL methanol extract or methanol was added to the agar-well, and methanol was used as the control.’ Thank you. (Please see page 5, lines 27-28).
Page 4, lines 32-33: Please revise this sentence. It does not read well.
We have changed the sentence to ‘Sporangia consisted of cylindrical and rough surfaced spores (0.6-0.8μm × 0.9-1.6μm) were produced on aerial mycelia, but spore chains were not observed.’ Thank you. (Please see page 5, lines 47-48)
Page 4, line 40: and also in Table S1?
We have changed the sentence to ‘Detailed physiological characteristics are presented in the species description, Table 2 and Table S1.’ Thank you. (Please see page 6, line 7).
Page 6, lines 11-1-33: In my view this text belongs in the Material and Methods section
We have moved the text to ‘Material and Methods’ section of 2.4 (2.4. Phylogenetic Analysis of NEAU-SSA 1T) and changed ‘Table 2’ to ‘Table 1’ and ‘Table 1’ to ‘Table 2’. Thank you. (Please see page 3, lines 25-32 and page 4, Table 1).
Page 7, lines 4-5: Please remove this sentence ‘The DNA G+C content of strain NEAU-SSA 1T was 69.9 mol%.’ It is repeated below and reads better there.
We have revised. Thank you. (Please see page 7, line 26)
Page 11, lines 35-36: Did the authors test other pathogenic bacteria? If yes, then this is “data not shown”?’
We evaluated the antibacterial activity of strain NEAU-SSA 1T against pathogenic bacterium of Micrococcus luteus, but no inhibitory effect was observed. We have showed the figure as the Fig. 4b. Thank you. (Please see page 11, lines 35-36 and Fig. 4b)
Page 11, line 45: If I am reading and understanding the idea correctly then the comma needs to be removed.
We have revised. Thank you. (Please see page 11, line 46)
Page 12, line 4: ‘and possesses 24 lower similarities’ gene clusters.’ related to putative antibacterial activity?
Thank you very much for your valuable suggestion. Based on the antiSMASH analysis, the strain contains 49 gene clusters, including 24 gene clusters, which showed very low similarity to the known gene clusters, these 24 gene clusters may relate to putative antibacterial activity or not. In this section, the sentence means that the strain may contain some new gene clusters and produce new substance. The strain is worth of studying in further research.
Page 12, line 9: Change ‘th’ to ‘the’
We have revised. Thank you. (Please see page 12, line 11)